# The outcomes of lockdown in the higher education sector during the COVID-19 pandemic

Peiling Cai[1]☯, Peng Ye[1]☯, Yihao Zhang[1], Rui Dai[1], Jingchun Fan [2], Brett D. Hambly[3], Shisan Bao [3]*, Kun Tao[3]*

1 School of Preclinical Medicine, Chengdu University, Chengdu, China, 2 School of Public Health, Centre for Evidence-Based Medicine, Gansu University of Chinese Medicine, Lanzhou, Gansu, China, 3 Department of Pathology, Tongji Hospital, School of Medicine, Tongji University, Shanghai, China

☯ These authors contributed equally to this work.
* taokun20119@163.com (KT); profbao@hotmail.com (SB)

## Abstract

To control COVID-19 pandemic, complete lockdown was initiated in 2020. We investigated the impact of lockdown on tertiary-level academic performance, by comparing educational outcomes amongst first-year students during second semester of their medical course prior to and during lockdown. *Evidence*: The demographics, including educational outcomes of the two groups were not significantly different during semester one (prior to the lockdown). The academic performance amongst women was better than men prior to lockdown. However, the scores were improved significantly for both sexes during lockdown in 2020, following the complete online teaching, compared to that in 2019, showing no significant difference between men and women in 2020, for English and Chinese History. There were significant different scores between men and women in lab-based Histology Practice in 2019 (in-person tuition) and 2020 (online digital tuition), although only a significant improvement in women was observed between 2019 and 2020. *Implication*: the forced change to online delivery of the second semester of the first-year medical program in 2020 due to the COVID-19 pandemic did not result in any decline in assessment outcomes in any of the subjects undertaken. We believe extensive online digital media should continue to be available to students in future.

## Introduction

Following the emergence of COVID-19 infections in China in December 2019, transmission rapidly occurred around the world [1, 2], resulting in a pandemic with severe social and economic impacts, with an unacceptably high mortality and morbidity [3].

Particularly effective management was instigated within China to control this emerging threat, including lockdowns, social distancing, minimisation of unnecessary activities [1, 2] and, importantly, the development of effective vaccines [4]. However, further challenges to the control of the pandemic have developed, including the emergence of the highly infectious and deadly delta and the most recent omicron strain of SARS-COV-2, which originally emerged

**Data Availability Statement:** All relevant data are within the paper and its Supporting Information files.

**Funding:** This research was supported by grants from First-class Curriculum Project of Chengdu

University in 2020 and 2021 (CDYLKC2021063 and CDYLKC2020016), awarded to PC. A grant from Teaching Reform Projects of Chengdu University in 2021 (cdjgb2021050), awarded to PC. Shanghai Jiaotong University Translational Foundation major project: Using single cell sequence, the specific target(s) of IL-38 in the precision medicine for colorectal cancer (ZH2018ZDA33, 2018-2021), awarded to KT. Shanghai Changning District Project for the Combination of Family Medicine and Community Doctor (YZJH003, 2020), awarded to KT. A grant from The Foundation of the Centre for Clinical Training, Shanghai Jiaotong University, awarded to KT. Teaching Achievement Cultivation Project of Gansu Province, grant no. 3136170103, awarded to JF. The 4th batch of Ideological and Political Demonstration Courses of Gansu University of Chinese Medicine- Evidence-based Medicine, Grant no. 313106020101, awarded to JF. The funders had no role in study design, data collection and analysis, decision to publish, or preparation of the manuscript.

**Competing interests:** The authors have declared that no competing interests exist.

from South Africa [5]. Despite effective and largely complete control of COVID-19 being achieved in China by June 2020, the δ strain of SARS-COV-2 emerged from overseas travellers in Nanjing in September 2021, with rapid spread to Yangzhou, Jiangsu Province [6, 7], and subsequent further spread to another 15 Provinces within China, resulting in severe physical and psychological pressures [7, 8]. Fortunately, the rapid and robust implementation of established pandemic control measures by Chinese authorities was able to control the δ strain outbreak within 4 weeks, supported by recent data showing a substantial reduction in COVID-19 transmissibility following the implementations of restrictions on mobility [9].

In response to the imposition of COVID-19-mandated restrictions requiring social distancing and limitations on mobility, extensive use of online training, mainly *via* Zoom, has been widely implemented, for example, within the health care industry, the development of an effective online training module for the clinical management of COVID-19 for Family Medicine and General Practitioners [10]. While the feasibility of online learning for medical students has been demonstrated, the overall outcomes of the switch to online learning remains to be explored [11].

All Chinese schools and universities were mandatorily closed during the first and second waves of the COVID-19 outbreak [2]. The responses applied by various educational institutions to the COVID-19 challenge varied [12], for example, transitioning from face-to-face teaching to online teaching whenever possible. However, from the point of view of higher education institutions, the immediate question is: what has been the impact for the students following the switch to mainly online teaching, particularly in relation to educational outcomes/performance?

Online education has been extensively and effectively adopted in China during the pandemic and has continued most recently in response to the resurgence of the δ strain of COVID-19 [7]. In the context of medical education, the impact of the switch to online teaching has not yet been evaluated, where the need for face-to-face interactions and hands-on practical laboratory-based experience are required, for example, in microscopy-histology practicals. In this study we sought to determine the effects of COVID-19-related changes in teaching on the performance of first-year medical students in the School of Preclinical Medicine, Chengdu University, China.

## Methods

Data concerning the demographics and educational performance for first-year medical students enrolled in 2018 (the pre-COVID-19 cohort), compared to those enrolled in 2019 (the COVID-19 cohort), were obtained from the electronic data system, Chengdu University School of Preclinical Medicine, Chengdu, China. Analysis and reporting of the de-identified data were approved by the Ethics Committee, Chengdu University School of Preclinical Medicine, and the consent is waived by the Ethics Committee. No of the students included in the current study was minor.

The medical course at Chengdu University is a 5-year program. The first 2 years of the program consists of Foundation Medical Studies, followed by 2 years of Clinical Studies and one year of Internship. The first semester of the program (Autumn semester) occurs between September and January each year, followed by the second semester (Spring semester) from March to July. Thus, the pre-COVID-19 cohort undertook their first semester of studies between September 2018 and January 2019, while the COVID-19 cohort undertook their first semester of studies between September 2019 and January 2020, immediately prior to the onset of the pandemic and consequent lockdown between February and July 2020 (Fig 1). Consequently, the first semester of study for both cohorts was unaffected by the COVID-19 pandemic and

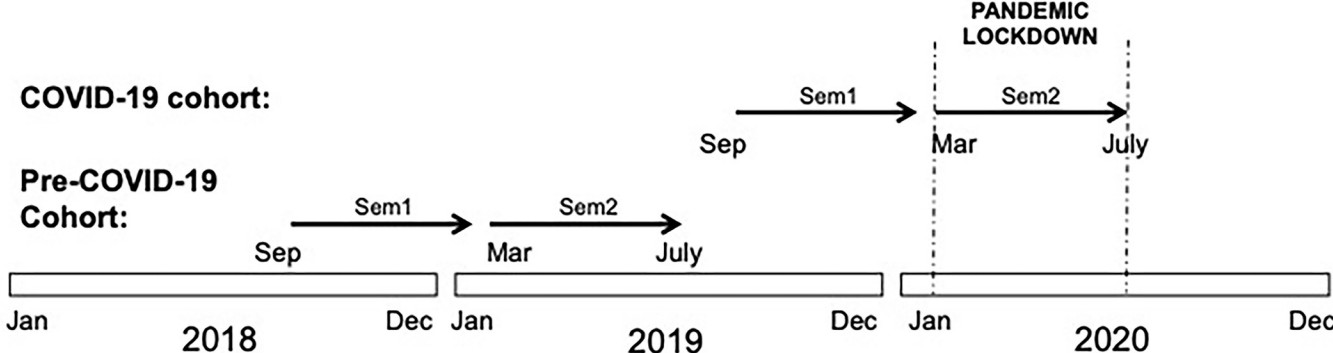

**Fig 1. Diagrammatic illustration of the experimental design sequence.** Chinese education consists of two semesters each year. Semester 1 includes September to the following February. Semester 2 March to July. Thus the COVID-19 cohort undertook the second semester of their medical course during pandemic lockdown between March and July 2020.

lockdown. However, while the pre-COVID-19 cohort undertook their second semester during March to July 2019, prior to the pandemic, the COVID-19 cohort undertook their second semester during the first COVID-19 lockdown from March to July 2020. Thus, to validate the equivalence of the two cohorts, educational outcomes specific to the two cohorts was evaluated prior to further analysis, by comparison of educational outcomes for first semester medical studies undertaken by the two cohorts prior to the onset of the pandemic. The first semester studies undertaken by the two cohorts were: Cell Biology, Advanced Mathematics, Chemistry and Medical Physics.

Eight pre-clinical subjects were undertaken during the second semester of the medical program (March to July each year): English, Computer Science, Chinese History, Sports, Biochemistry, Anatomy, Histology and Histology Practice. The pre-COVID-19 cohort studied these subjects between March and July 2019, prior to the pandemic, while the COVID-19 cohort undertook semester 2 during March to July 2020, during the COVID-19 pandemic and lockdown. Educational outcomes between the two cohorts were evaluate for each cohort as a whole and further stratified by sex.

Notably, the content of four of these eight subjects (English, Computer Science, Chinese History and Biochemistry) was presented entirely by lecture format in both 2019 and 2020. In 2019 these lectures were presented in a timetabled face-to-face format, with only the Powerpoint content of the lecture being made available to the students online. However, in 2020 the pandemic mandated that lectures in these subjects were lived-streamed online to all students at a timetabled fixed time. Additionally, during 2020, all these lectures were also specifically recorded and made available online by the university authorities, allowing students to subsequently review the lecture content as often as required.

On the other hand, prior to the pandemic, Sports was a subject that utilised an extensive face-to-face practical methodology. However, to teach and assess this subject in 2020, tuition for students was provided online, and students were asked to send in a video clip to demonstrate their physical activities and achievements.

Importantly, both Anatomy and Histology consist of both lectures and laboratory-based practical sessions. Anatomy tuition was split 50% each between lectures (48 hours) plus in-person practical sessions (48 hours). In 2019, these two parts were carried out separately: in-person timetabled lectures and in-person timetabled laboratory practical sessions. However, in 2020, face-to-face lectures were replaced by live-streamed online lectures, meanwhile, personal practical sessions in the laboratory were replaced by watching experimental operation videos

online. The practical component of Anatomy was incorporated into the overall assessment of the Anatomy subject. However, in the case of Histology and Histology Practical, Histology Practical, was taught and assessed separately to the Histology subject, of which the latter consisted entirely of lecture-style teaching. However, notably, educational innovation was introduced into Histology in 2018 by the introduction of the "flipped classroom" teaching methodology, which aimed to increase student engagement and learning by providing the required learning material to students prior to the in-person learning sessions, both written and lecture-style pre-recorded media. The in-person learning sessions that replaced the previous lecture format were then focused on live problem-solving and addressing questions from the students.

In the case of Histology Practice, during 2019 students attended timetabled Practical classes and used microscopes and glass microscopic slides to view tissue under study, with reasonable access to revision of this microscopic material being available during subsequent schedules revision sessions conducted in the practical laboratory. Powerpoints and other written material for this subject were available to students online during 2019. However, during 2020, all teaching activities were conducted online and all material was accessed online, including written material and access to digital, scanned interactive microscopic slide images, that incorporate the entire microscopic slide, with the capacity to vary the magnification and plane of focus as required, thus allowing students to view and review all parts of the microscopic slide at any time, in a similar manner to using a physical microscope and microscopic slide.

Furthermore, specifically in the case of Histology Practice, to determine if there was an impact of COVID-19 restrictions on the students' performance in hands-on microscope-based teaching, comparisons were undertaken on a range of assessments, namely: daily performance, the mid-term examination, the final examination and overall scores for the first-year medical students, for the cohort in 2019 compared to 2020, including a comparison of men vs women students.

In all subjects, the overall assessment resulting in the final grades consisted of three elements: the Daily Performance, Mid-term Tests and the Final Test, which followed the criteria for teaching management at Chengdu University. Thus, the final grades reflected the overall performance of the students throughout each term. In addition, as an example, for the subject Histology Practice we have provided detailed data on these three specific elements by applying The Electronic Teaching Management System (Fig 4), to acquire more detailed academic performance data and carry out a more adequate statistical analysis.

## Statistical analysis

The continuous variables that conforming to a normal distribution were described by the mean and standard deviation ($\bar{x} \pm s$), and the independent sample $t$ test (homogeneous variance) or $t'$ test (uneven variance) were used for comparison between the two groups. The Kolmogorov-Smirnov test was used to determine whether the scores of each course obeyed a normal distribution and to compare the two datasets. For quantitative data that did not obey a normal distribution, that is, that exhibited a skewed distribution, the data were represented by the median and interquartile range, and the Mann-Whitney U test was used to examine the association between the two participant groups. Using GraphPad Prism 9.0, all statistical analysis was based on a two-sided hypothesis test, and the test level was set to $\alpha = 0.05$.

## Results and discussion

### Basic students' number, age and sex

There was no significant difference in the total number of students enrolled in Medicine 1 between the 2018 pre-COVID-19 cohort and the 2019 COVID-19 cohort (82 vs 91 students) (Fig 2A).

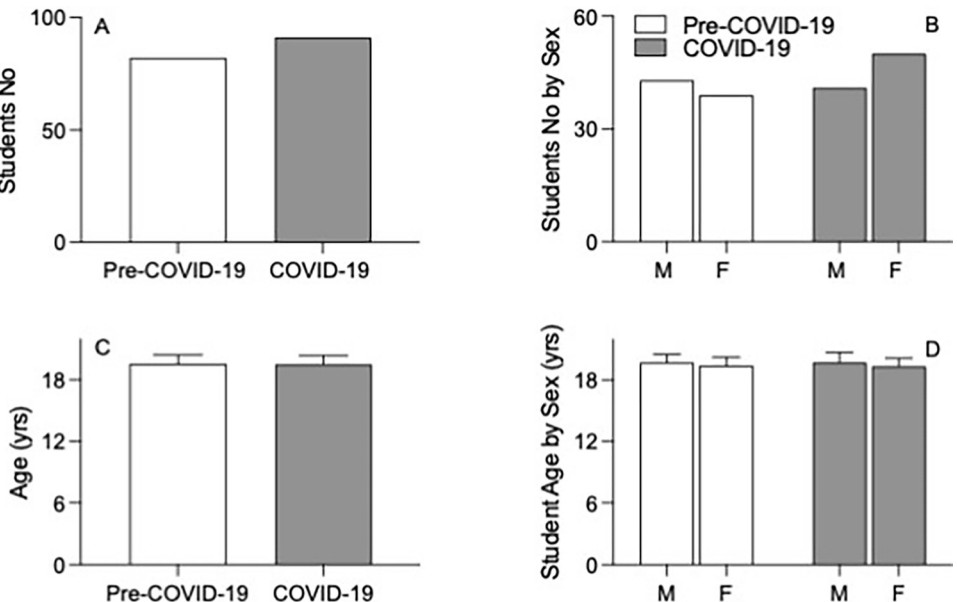

**Fig 2.** The comparison between the students' demographics for the pre-COVID-19 cohort (white bars) and COVID-19 cohort (grey bars) (A), students' numbers stratified by sex (B), the average age of the students (C) and the students' age stratified by sex (D).

Similarly, while there was a slightly lower number of women students between 2018 and 2019 (39 vs 50 students), compared to that of the men students (43 vs 41 students) (Fig 2B), the difference was not significant. Thus, there were no significant differences among these four groups.

Meanwhile, there was no significant differences in the average age of student between the 2018 pre-COVID-19 cohort and the 2019 COVID-19 cohort ($19.57 \pm 0.85$ vs $19.48 \pm 0.87$ years) (Fig 2C). Similarly, while there was a slightly lower average age of men students between 2018 and 2019 ($19.74 \pm 0.79$ vs $19.66 \pm 0.97$ years), compared to that of the women students ($19.38 \pm 0.88$ vs

$19.34 \pm 0.80$ years) (Fig 2D), the differences were not significant. Consequently, there were no significant differences among these four groups.

The mean age of students in 2018 was $19.57 \pm 0.85$ years, compared to 2019 of $19.48 \pm 0.87$ years. There was no statistical difference in the overall mean age between the two groups (difference [d] = 0.09, 95% confidence interval [95% CI] -0.17–0.35, $t = 0.678$, $P = 0.498$) (Fig 1C). Furthermore, the students were further divided into subgroups by sex. For 2018, the mean age of men students was $19.74 \pm 0.79$ years, while the mean age of women students was $19.38 \pm 0.88$ years; for 2019, the mean age of men was $19.66 \pm 0.97$ years, the mean age of women was $19.34 \pm 0.80$ years. There was no significant difference among these four groups (men-2018 vs women-2018: d = 0.36, 95% CI -0.01–0.73, $t = 1.954$, $P = 0.054$; men-2019 vs women-2019: d = 0.32, 95% CI -0.05–0.69, $t = 1.724$, $P = 0.088$; men-2018 vs men-2019: d = 0.09, 95% CI -0.30–0.47, $t = 0.446$, $P = 0.657$; women-2018 vs women-2019: d = 0.05, 95% CI -0.31–0.40, $t = 0.251$, $P = 0.803$) (Fig 1D).

## Baseline comparability for academic performance undertaken during the first semester of medical school and prior to the COVID-19 pandemic

Both cohorts undertook their first semester of medical school prior to the COVID-19 pandemic. The pre-COVID-19 cohort undertook their pre-medical studies during 2018 and the

COVID-19 cohort during 2019. With the exception of Chemistry, there was no significant difference in educational outcomes between these two student cohorts, when analysed using the first semester pre-medical subjects: Cell Biology, Advanced Mathematics, Chemistry and Medical Physics (S1 Fig). In the case of Chemistry, the educational outcomes for the COVID-19 cohort were slightly lower than for the pre-COVID-19 cohort (76.39 ± 6.68; $P$ = 0.002), with the difference persisting when the students were stratified by sex. Overall, these data support the conclusion that the cohorts are equivalent in their inherent academic ability, hence supporting the subsequent comparisons as being valid.

## Second semester medical school learning: Comparison between pre-COVID-19 face-to -face teaching and the COVID-19 online format

When student performance in English was examined, it was observed that women student performance was 1.1-fold higher than men performance within the pre-COVID-19 cohort ($P$ = 0.0002) (Fig 3A). Two changes in student performance in English were observed when the pre-COVID-19 cohort, who received their lectures face-to-face, was compared to the COVID-19 cohort, who received their lecture content via online streaming (Fig 3A). Firstly, the performance of both men and women students improved between 2019 and 2020, by a factor of 1.2-fold ($P<0.0001$) and 1.1 fold ($P<0.0001$), respectively. Secondly, the larger improvement for the men cohort compared to the women cohort between 2019 and 2020, resulting in there being no difference in performance between the sexes in 2020 ($P$ = 0.469).

A very similar pattern to that observed for English performance was observed in the performance in Chinese History (compare Fig 3A to 3C), namely in 2019, women performed better than men. However, both men and women performance was improved in 2020, with no difference being observed between the sexes in 2020.

The performance in the three subjects Biochemistry, Anatomy and Histology also partly followed a similar pattern to the performance in English and Chinese History. Specifically, in each of these subjects, the performance of women students was superior to men students in 2019 (Biochemistry $P$ = 0.0003, Anatomy $P$ = 0.0002, Histology $P$ = 0.0008) (Fig 3E–3G). Similarly, there was an improvement in men performance between 2019 and 2020 in Biochemistry ($P$ = 0.001) and Anatomy ($P$ = 0.037), but not in Histology ($P$ = 0.272). However, in these three subjects, women performance did not significantly improve between 2019 and 2020. Interestingly, in two subjects, Anatomy and Histology, there was no difference in performance between the sexes in 2020, but the superior performance of women compared to men persisted in Biochemistry in 2020 ($P$ = 0.028).

On the other hand, there was no significant difference in Computer Science scores between men and women in both 2019 and 2020 (Fig 3B). However, women in 2020 improved their Computer Science scores both in comparison to women in 2019 (1.2-fold improvement; $P$ = 0.009) and men in 2019 (1.1-fold improvement; $P$ = 0.003).

The subject of Sports (Physical exercise) was undertaken in 2019 utilising in-person tuition. However, to teach and assess the subject in 2020 during lockdown, tuition for students was provided online, and students were asked to send in a video clip to demonstrate their physical activities and achievements, which may have had the potential to lead to bias in the assessment. A comparison of assessment outcomes between 2019 (in-person training) versus 2020 (online tuition and assessment) reveals a significant improvement in scores for both men ($P$ = 0.004) and women ($P<0.0001$) students (Fig 3D). However, there was no difference in scores between men and women specifically in either 2019 or 2020.

The teaching of Histology Practice has always been heavily-dependant on in-person tuition, within a microscope laboratory utilising glass microscope slides. However, in 2020 the

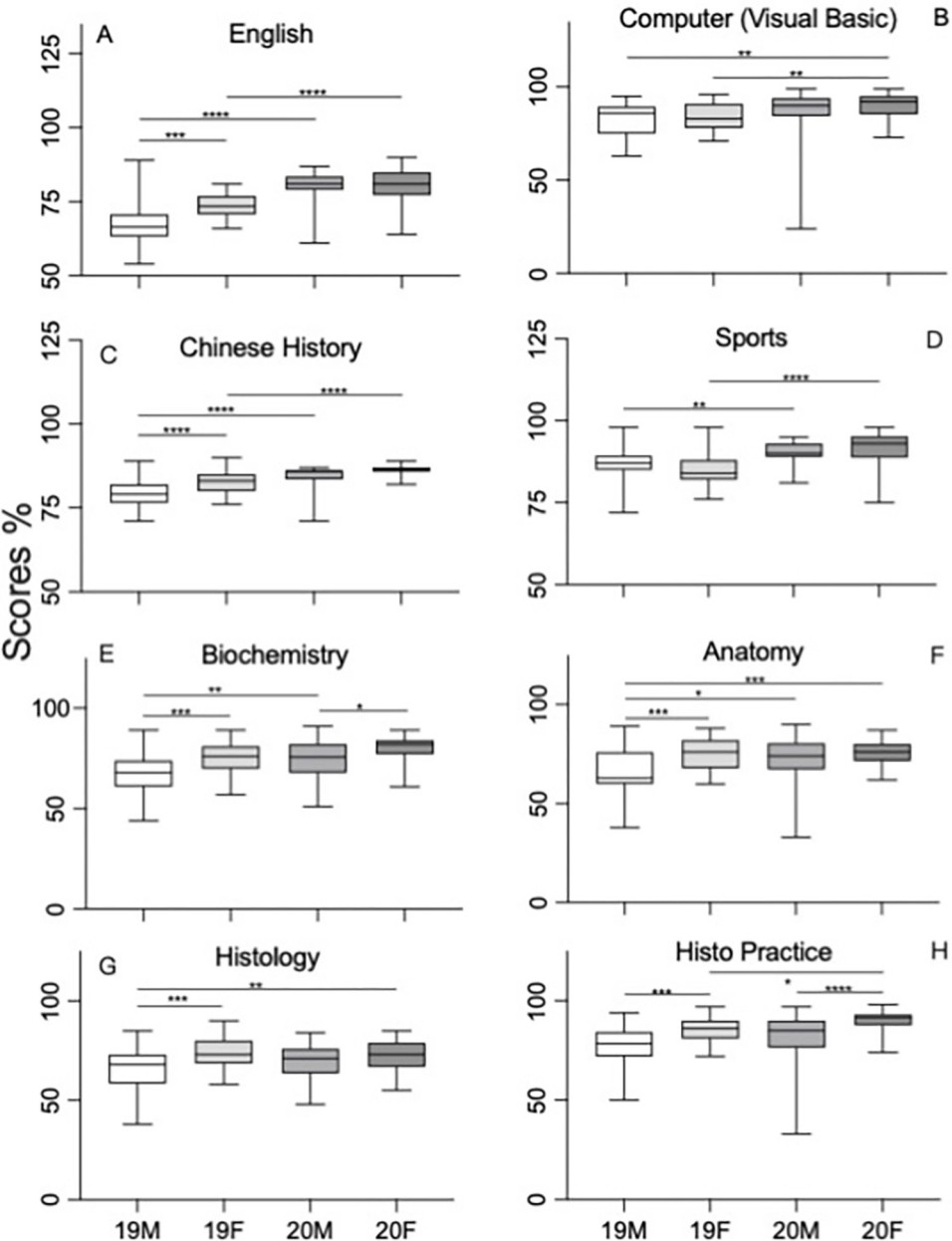

**Fig 3.** Student academic performance by subject, English (A), Computer Science (B), Chinese History (C), Sports (D), Biochemistry (E), Anatomy (F), Histology (G) and Histology Practice (H). The white, light grey, middle grey and dark grey bars represent the men student in 2019, women student in 2019, men students in 2020 and women students in 2020, respectively. The Y-axis represents the mark in percentage points.

requirement to teach the subject online necessitated the transition of Histology Practice teaching to the use of only online teaching, including the use of digital microscopic slides. Notably, the performance of women students in this subject was found to be superior to men in both 2019 ($P = 0.0008$) and 2020 ($P<0.0001$) (Fig 3H). Additionally, between 2019 and 2020, the performance of women students was found to have improved ($P = 0.046$), while the performance of men students did not improve.

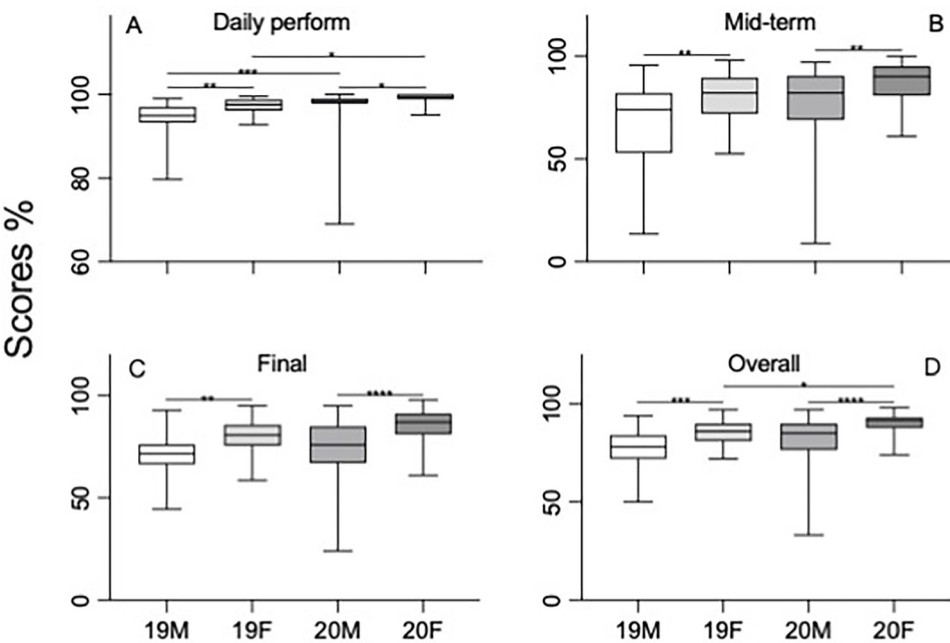

**Fig 4. The academic performance achieved during progressive assessment in histology practice.** Daily performance mark (A), mid-term examination mark (B), final examination mark (C) and overall mark (D) are shown. The white, light grey, middle grey and dark grey bars represent the men student in 2019, women student in 2019, men students in 2020 and women students in 2020, respectively. The Y-axis represents the mark in percentage points.

To further understand the overall assessment outcomes for Histology Practice, the assessment outcomes of the individual components contributing to the overall assessment were evaluated (Fig 4). In all assessment tasks in both 2019 and 2020, women achieved superior scores to men. However, there was only an improvement in the Daily Performance scores between 2019 and 2020 (Fig 4A), for both men ($P = 0.0004$) and women ($P = 0.023$), with no differences being observed between the scores for 2019 and 2020 in either the Mid-term (Fig 4B) or Final (Fig 4C) assessments.

The severity of the COVID-19 pandemic has extensively affected normal life, including a substantial impact on higher education learning. The Minister for Education in China specifically mandated the requirement for changes in university teaching, i.e. "suspending classes without suspending learning", to comply with the broader directive to social distance/lockdown communities in response to the pandemic. Thus, in 2020 all university tuition was moved from a conventional teaching format (in-person lectures) to online teaching [13]. However, inevitably the restrictions inherent in social distancing/lockdowns have led to a range of consequences of differing severity, such as post-traumatic stress symptoms and anxiety [14].

The aim of this study was to determine the extent to which the COVID-19 pandemic impacted learning outcomes for a cohort of first year medical students, noting that the major changes caused by the COVID-19 pandemic for this group related to both general restrictions mandated by the need for social distancing/lockdown, and, specifically, the immediate move to online tuition during 2020.

A necessary pre-requisite to this study was the need to demonstrate that the two cohorts being compared were sufficiently similar to each other. This was achieved by examining the educational outcomes for the first semester of pre-medical studies undertaken by the two cohorts prior to the onset of the pandemic, in the case of the pre-COVID-19 cohort during

2018, and in the case of the COVID-19 cohort, during 2019. These data showed that there were no significant differences in these two cohorts in terms of the total student numbers, which is largely pre-determined by the university administration. Additionally, there was no significant differences in student age, largely due to the majority of students being recruited directly from high school. However, there was a slightly higher, but non-significant, number of women students compared to men students within both cohorts, most likely related to a preference by women to become medical practitioners. Importantly, when academic outcomes were examined for first semester pre-medical studies undertaken prior to the COVID-19 pandemic, essentially no differences were observed, with the exception of Chemistry, where only a minimal difference was observed. Hence, these data support the conclusion that the two cohorts were equivalent in their inherent academic ability, validating subsequent comparisons.

When academic outcomes were examined between the two cohorts during the second semester of medical school, comparing the pre-COVID-19 cohort to the COVID-19 cohort, several clear observations emerged. Firstly, in the majority of subjects, women students performed better than men students both pre-COVID-19 and during the pandemic. Academic performance can be influenced by factors such as self-motivation and conscientiousness [15]. The better performance by women in tertiary medical courses has been demonstrated previously by Wu *et al.* (2020) [16], who ascribed this difference to a higher level of self-efficacy amongst women students, i.e. that women students subjectively perceive that they possess a higher level of confidence that they were capable to complete the learning tasks required within their course of study, leading to greater mental effort and persistence, compared to men students. By contrast, a study of emergency competencies amongst resident doctors-in-training found that men were rated as performing slightly better than women in 4 out of 22 patient care sub-competencies that were assessed, although by end of training this small difference had disappeared [17]. Notably, the resident doctors within this study were at the end of their training, so were older and consequently more mature and likely more responsible, compared to the first-year medical students described in the present study [18].

The second observation that emerged from this study was that for most subjects, compared to the pre-COVID-19 cohort, the academic scores achieved by both men and women students within the COVID-19 cohort were significantly improved. One possible explanation for this outcome is that students within the COVID-19 cohort were subjected to substantial constraints on potential distractions to their learning due to social distancing/lockdown, severely limiting their capacity to engage in distractions that would be expected to include social and entertainment activities [1, 2]. Thus, we propose that lockdown encouraged students to focus on their academic duties and spend more time on their academic learning.

Importantly, tuition was delivered entirely online for the COVID-19 cohort. While there was initial concern that the sudden move of tuition from face-to-face learning to an online format may detrimentally affect learning outcomes, the data from this study demonstrate that, fortunately, the opposite outcome was achieved, namely that there was either no change or, more often, an improvement in assessment outcomes across all subjects, with the improvement being more marked in some subjects compared to others. An OECD policy brief published in 2005 concluded that in tertiary education, online resources may be used to either supplement a course, present a mixed-mode course dependent on at least some e-learning, or present a fully online course (E-learning in Tertiary Education, OECD). Each of these options carries advantages and disadvantages, although at that time (2005), the dominant form of e-learning continued to be supplementary to on-campus delivery, usually delivered through purpose-built Learning Management Systems (LMS), that generally include online access to course materials such as recordings of lectures, notes and other media. Advantages cited for e-learning include flexibility of access and increased capacity for detailed review of learning

materials at the student's own pace. Thus, an additional possible explanation for the improved assessment outcomes seen in this study may be the sudden introduction of extensive online access for students to e-learning resources for the COVID-19 cohort of students, as a consequence of the live-streaming of lectures, and subsequent access to digital recordings for detailed revision of the learning material.

On the other hand, careful consideration must be given to the long term complications of lockdown from a psychological point of view, as has been demonstrated by an increase in psychological disorders among patients visiting a primary care practice [19] and amongst university students [20]. These data are consistent with other work showing that restrictive lockdowns result in varying severity of post-traumatic stress symptoms and anxiety [14].

The assessment outcomes in Computer Science differed slightly from that seen more generally for other subjects. Specifically, it was found that there was no difference in assessment outcomes between men and women prior to the pandemic, but that, as a consequence of the switch to online teaching during the pandemic, the performance of women was improved, but there was no change in the performance of the men. Our data are supported by Milesi et al. (2017) [21], who showed that women students achieve to the same level as men students in Computer Science, although there are obstacles associated with under-representation of women students in Computer Science, perhaps leading to enhanced learning amongst women students *via* persistence and engagement. On the other hand, there was an improvement in assessment outcomes for women students as a consequence of the switch to online tuition among the COVID-19 cohort. A possible explanation for this outcome may be that women students were able to benefit from the increased time available to them due to lockdown, allowing them to harness greater persistence and engagement in their computer science studies. Additionally, it has been observed that men students are generally less stressed by their learning in Computer Science, compared to women [21], with or without COVID-19.

A notable observation is that the improvements observed in the two Humanities subjects, namely English and Chinese History, were almost identical. Firstly, there was a significantly higher assessment outcome for women compared to men in the pre-COVID-19 cohort, secondly, there was a significant improvement amongst both men and women for the COVID-19 cohort and, thirdly, there was no significant difference between men and women within the COVID-19 cohort, i.e. the improvement in the performance of the men was larger than that amongst the women. Overall, this outcome is particularly important, given that there was initial concern that the move to online teaching in 2020 during the COVID-19 pandemic may be detrimental to assessment results, which fortunately was avoided. This successful outcome is also in line with the Chinese higher education policy, namely "suspending classes, without suspending learning" [13].

In contrast, oncomes in Biochemistry, Anatomy and Histology subjects exhibited slight variations on the pattern of improvement seen with English and Chinese History. In these three subjects, only an improvement in the man scores was observed in the COVID-19 cohort in 2020, which elevated the man scores up to the level of the women scores in 2020, compared to 2019. These data are consistent with women performance in these three subjects already being at a high level in 2019, with no room for further improvement in 2020. These data are consistent with the hypothesis that even before the pandemic, women students were sufficiently motivated to study and achieve at a high level, irrespective of the mode of tuition delivery, while man students were more likely to benefit from the additional time available to them as a consequence of lockdown, plus the opportunity for enhanced revision of learning material as a consequence of educational media availability online in digital format. These hypotheses are consistent with previous data that show that women students are able to achieve higher assessment results in complementary and alternative medicine, compared to men [22].

The improved assessment outcomes for Sports (Physical Exercise) during the pandemic in 2020 were a surprise, given that there was a complete lockdown for 4 months of 2020, that was only gradually relaxed in the latter half of 2020. The urgent need to implement some form of online scoring system in 2020 may have led to a sub-optimal and subjective assessment, where students were asked to self-record their "mandatory" sports activities and return video clips of their sporting achievements within their student residences for scoring. While the experience of 2020 may lead to modifications in tuition and assessment in the future, that involve more innovative and objective assessment methodologies, hopefully these will not be needed if further lockdowns can be avoided.

Histology Practice assessments yielded some interesting data that differed slightly from the other subjects examined. In both 2019 and 2020, assessment outcomes for women were superior to men. However, only women improved their assessment outcomes in 2020 compared to 2019. The reasons for the failure of the man students to improve their educational outcomes in 2020 are unclear.

Finally, to further investigate the assessment outcomes in depth within the Histology Practice subject, we examined outcomes for each of the three assessment tasks that students undertook, namely, daily performance, the mid-term assessment and the final assessment. The usual pattern observed for the other subjects was observed for the daily performance, characterised by a superior women performance in both 2019 and 2020, with an improvement in performance for both men and women between 2019 and 2020. A possible explanation for this improvement between 2019 and 2020 may relate to improved access to brief online revision for students in 2020 prior to the daily online assessment. However, no improvement in performance was observed between 2019 and 2020 for either the mid-term or final assessment. A possible explanation for this outcome is that students already had access to both substantial online learning resources and laboratory-based revision classes in 2019, with similar access in 2020, albeit that revision was conducted online using digitised microscopic slides in 2020, resulting in no further improvement in assessment outcomes in 2020.

The study was focusing on the second semester of the first-year medical course, which covers foundational studies. However, notably the courses included histology via lectures and practicals, which offers a bridging course for the fresh students gradually moving from being laypersons towards medically oriented study. Despite many of the courses selected in the current study being pre-clinical, the aim of the current study was exploring the impact of the COVID-19 pandemic, particularly lockdown, in higher education, amongst first year medical students. Our data offers rather convincing evidence that digital education is a reliable approach for the students, compared to the conventional face-to-face teaching. In a future study, we will extend the cohort into senior medical students to further explore the impact of lockdown restrictions on clinical academic performance.

Limitation: The students were the first year medical students only, it would be good to extend the information to other years, particularly in year 5 (internship) for further understanding the impact of COVID-19 in higher education. In addition, using a larger cohort with different institutions from different provinces would also be useful to obtain an overall picture of the situation, which will be determined in future.

Notably, it is possible that the differences in performance that we observed may be due to certain difference in the content that was provided to students and/or differences in the difficulty of the assessments that were undertaken. With the possible exception of Sports, the teaching staff involved were confident that both the content and assessments were very close to identical between the two cohorts. However, our future studies will verify this point.

Additionally, a useful insight may be gained in future by surveying data concerning the students' self-assessments perception about the extent of their mastery of some specific essential knowledge in each subject, which will be performed in a future study.

## Conclusions

The forced change to online delivery of the second semester of the first-year medical program in 2020 due to the COVID-19 pandemic did not result in any decline in assessment outcomes in any of the subjects undertaken. On the contrary, for the majority of the subjects undertaken, the transition to online classes resulted in an improvement in assessment outcomes. This improvement may be attributed to two potential factors, firstly, the greater availability of time to dedicate to learning as a result of lockdowns, and, secondly, due to the ability of students to engage in online revision of the expanded digital teaching media available to them as a result of the transition to online teaching. These data support the recommendation that extensive online digital media should continue to be available to students in future, irrespective of their return to more formal face-to-face teaching, to improve overall assessment outcomes moving forward. An additional conclusion from these data is that within the age group studied (approximately 19 years of age) women students consistently perform better than man students, possibly as a consequence of the higher level of maturity, and hence motivation, of women within this age cohort.

## Supporting information

**S1 Fig.** Validation of the pre-pandemic academic performance between the pre-COVID-19 cohort and COVID-19 cohort for the semester 1 studies undertaken prior to the pandemic on the subjects: Cell Biology, Advanced Mathematics, Chemistry and Medical Physics (A). The Y-axis represents the mark in percentage points. Validation of the marks achieved following stratification by sex for the two cohorts (B).
(DOCX)

**S1 Data.**
(PDF)

## Author Contributions

**Conceptualization:** Peiling Cai, Shisan Bao, Kun Tao.

**Data curation:** Peng Ye, Rui Dai, Jingchun Fan, Kun Tao.

**Formal analysis:** Peng Ye.

**Methodology:** Peiling Cai, Yihao Zhang.

**Validation:** Brett D. Hambly, Kun Tao.

**Writing – original draft:** Peiling Cai, Shisan Bao.

**Writing – review & editing:** Peiling Cai, Brett D. Hambly, Shisan Bao, Kun Tao.

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
