## [Decision Letter · Decision Letter 0]

8 Dec 2022

PONE-D-22-27104The outcomes of lockdown in the higher education sector during the COVID-19 pandemicPLOS ONE

Dear Dr. Bao,

Thank you for submitting your manuscript to PLOS ONE. After careful consideration, we feel that it has merit but does not fully meet PLOS ONE’s publication criteria as it currently stands. Therefore, we invite you to submit a revised version of the manuscript that addresses the points raised during the review process.

We look forward to receiving your revised manuscript.

Kind regards,

Xiangjie Kong

Academic Editor

PLOS ONE

Journal Requirements:

2. Please ensure that you have specified (1) whether consent was informed and (2) what type you obtained (for instance, written or verbal, and if verbal, how it was documented and witnessed). If your study included minors, state whether you obtained consent from parents or guardians. If the need for consent was waived by the ethics committee, please include this information.

3. Please change "female” or "male" to "woman” or "man" as appropriate, when used as a noun (see for instance https://apastyle.apa.org/style-grammar-guidelines/bias-free-language/gender).

   "nil"

6. We note that you have indicated that data from this study are available upon request. PLOS only allows data to be available upon request if there are legal or ethical restrictions on sharing data publicly. For more information on unacceptable data access restrictions, please see http://journals.plos.org/plosone/s/data-availability#loc-unacceptable-data-access-restrictions. 

Reviewers' comments:

Reviewer's Responses to Questions

**Comments to the Author**

1. Is the manuscript technically sound, and do the data support the conclusions?

Reviewer #1: Partly

Reviewer #2: Yes

2. Has the statistical analysis been performed appropriately and rigorously? 

Reviewer #1: Yes

Reviewer #2: Yes

3. Have the authors made all data underlying the findings in their manuscript fully available?

Reviewer #1: Yes

Reviewer #2: Yes

4. Is the manuscript presented in an intelligible fashion and written in standard English?

Reviewer #1: Yes

Reviewer #2: Yes

5. Review Comments to the Author

Reviewer #1: Authors in this paper investigated the impact of lockdown on tertiary-level academic performance, by comparing educational outcomes amongst first-year students during second semester of their medical course prior to and during lockdown. This paper reveals to us a very interesting conclusion: the scores were improved significantly for both sexes during lockdown in 2020, following the complete online teaching.

There some suggestions and concerns are listed as following:

1. Is it reasonable to use final grades to measure students' learning outputs? Is it due to certain differences in the content and difficulty of the exam?

2. Two specific examples are suggested to illustrate the plausibility of the results of this study, such as comparing learning outcomes by comparing the scores of two groups of students on computer programming questions with similar academic requirements.

3. It is recommended that the authors collect and study data of students' self-assessments about their mastery degree of some specific knowledge.

Reviewer #2: An interesting study that covered majority of the first year medical course subjects undertaken prior to and during COVID-19. I cannot find any fault to it since all have been explained, analysed and discussed very well.

6. PLOS authors have the option to publish the peer review history of their article (what does this mean?). If published, this will include your full peer review and any attached files.

Reviewer #1: No

Reviewer #2: No

---

## [Author Response · Author response to Decision Letter 0]

8 Feb 2023

Dr Xiangjie Kong

Academic Editor

PLOS ONE

PONE-D-22-27104

Dear Dr Kong

We appreciate the constructive comments made by the reviewers, our responses are as follows:

1. Is the manuscript technically sound, and do the data support the conclusions?

Reviewer #1: Partly

Reviewer #2: Yes

The manuscript is being revised accordingly to improve the quality as specified by reviewer #1 in the following sections.

2. Has the statistical analysis been performed appropriately and rigorously? 

Reviewer #1: Yes

Reviewer #2: Yes

Thanks 

3. Have the authors made all data underlying the findings in their manuscript fully available?

Reviewer #1: Yes

Reviewer #2: Yes

Thanks

4. Is the manuscript presented in an intelligible fashion and written in standard English?

Reviewer #1: Yes

Reviewer #2: Yes

Thanks 

5. Review Comments to the Author

Reviewer #1: This paper reveals to us a very interesting conclusion.

Thanks 

There some suggestions and concerns are listed as following:

1. a. Is it reasonable to use final grades to measure students' learning outputs? 

This point requires clarification, so we have added the following section in Materials and Methods, it now reads: 

“In all subjects, the overall assessment resulting in the final grades consisted of three elements: the Daily Performance, Mid term Tests and the Final Test, which followed the criteria for teaching management at Chengdu University. Thus, the final grades reflected the overall performance of the students throughout each term. In addition, as an example, for the subject Histology Practice we have provided detailed data on these three specific elements by applying The Electronic Teaching Management System (Fig. 4), to acquire more detailed academic performance data and carry out a more adequate statistical analysis.” (Materials and Methods).

b. Is it due to certain differences in the content and difficulty of the exam? 

(2) Agree, we have added the following section in the Discussion, it now reads: 

“Notably, it is possible that the differences in performance that we observed may be due to certain difference in the content that was provided to students and/or differences in the difficulty of the assessments that were undertaken. With the possible exception of Sports, the teaching staff involved were confident that both content and assessments were very close to identical between the two cohorts. However, our future studies will verify this point.” 

2. Two specific examples are suggested to illustrate the plausibility of the results of this study, such as comparing learning outcomes by comparing the scores of two groups of students on computer programming questions with similar academic requirements.

We agree completely. However, the current study was performed 2 years ago in response to a specific emergency situation (the COVID-19 pandemic lockdowns) and the teaching model that has been subsequently changed into a combined online and face-to-face teaching format, partly in response to the data presented in this manuscript. However, we will perform such testing in a future study.

3. It is recommended that the authors collect and study data of students' self-assessments about their mastery degree of some specific knowledge.

We appreciate such constructive comments, however, we are not able to pursue this question further, because the students have reached clinical years now, and their depth of clinical perspective in relation to the basics sciences has changed since 2020. Thus, the students’ self-assessments may not be objective or clear to enable the test of this hypothesis. We have added this limitation into the Discussion, it now reads: “Additionally, a useful insight may be gained in future by surveying data concerning the students' self-assessments perception about the extent of their mastery of some specific essential knowledge in each subject, which will be performed in a future study.”

Reviewer #2: An interesting study that covered majority of the first year medical course subjects undertaken prior to and during COVID-19. I cannot find any fault to it since all have been explained, analysed and discussed very well.

Thanks 

We have modified our manuscript accordingly. 

Yours sincerely

Shisan Bao

---

## [Decision Letter · Decision Letter 1]

27 Feb 2023

The outcomes of lockdown in the higher education sector during the COVID-19 pandemic

PONE-D-22-27104R1

Dear Dr. Bao,

We’re pleased to inform you that your manuscript has been judged scientifically suitable for publication and will be formally accepted for publication once it meets all outstanding technical requirements.

Kind regards,

Xiangjie Kong

Academic Editor

PLOS ONE

Additional Editor Comments (optional):

Reviewers' comments:

Reviewer's Responses to Questions

**Comments to the Author**

1. If the authors have adequately addressed your comments raised in a previous round of review and you feel that this manuscript is now acceptable for publication, you may indicate that here to bypass the “Comments to the Author” section, enter your conflict of interest statement in the “Confidential to Editor” section, and submit your "Accept" recommendation.

Reviewer #1: All comments have been addressed

2. Is the manuscript technically sound, and do the data support the conclusions?

Reviewer #1: Yes

3. Has the statistical analysis been performed appropriately and rigorously? 

Reviewer #1: Yes

4. Have the authors made all data underlying the findings in their manuscript fully available?

Reviewer #1: Yes

5. Is the manuscript presented in an intelligible fashion and written in standard English?

Reviewer #1: Yes

6. Review Comments to the Author

Reviewer #1: Authors in this paper investigated the impact of lockdown on tertiary-level academic performance, by comparing educational outcomes amongst first-year students during second semester of their medical course prior to and during lockdown. This paper reveals to us a very interesting conclusion: the scores were improved significantly for both sexes during lockdown in 2020, following the complete online teaching. The previous problems have been solved perfectly.

7. PLOS authors have the option to publish the peer review history of their article (what does this mean?). If published, this will include your full peer review and any attached files.

Reviewer #1: No

---

## [Editor Report · Acceptance letter]

14 Apr 2023

PONE-D-22-27104R1 

The outcomes of lockdown in the higher education sector during the COVID-19 pandemic 

Dear Dr. Bao:

I'm pleased to inform you that your manuscript has been deemed suitable for publication in PLOS ONE. Congratulations! Your manuscript is now with our production department. 

Kind regards, 

on behalf of

Dr. Xiangjie Kong 

Academic Editor

PLOS ONE